# Complete plastid genome of *Iris orchioides* and comparative analysis with 19 *Iris* plastomes

**Tae-Young Choi** [iD], **Soo-Rang Lee** [iD]*

Department of Biology Education, Chosun University, Gwangju, South Korea

* ra1130@hotmail.com, ra1130@chosun.ac.kr

**Data Availability Statement:** Raw genomic data used in the analyses are deposited in GenBank (https://www.ncbi.nlm.nih.gov/genbank/) with accession number, PP475539.

## Abstract

*Iris* is a cosmopolitan genus comprising approximately 280 species distributed throughout the Northern Hemisphere. Although *Iris* is the most diverse group in the Iridaceae, the number of taxa is debatable owing to various taxonomic issues. Plastid genomes have been widely used for phylogenetic research in plants; however, only limited number of plastid DNA markers are available for phylogenetic study of the *Iris*. To understand the genomic features of plastids within the genus, including its structural and genetic variation, we newly sequenced and analyzed the complete plastid genome of *I. orchioides* and compared it with those of 19 other *Iris* taxa. Potential plastid markers for phylogenetic research were identified by computing the sequence divergence and phylogenetic informativeness. We then tested the utility of the markers with the phylogenies inferred from the markers and whole-plastome data. The average size of the plastid genome was 152,926 bp, and the overall genomic content and organization were nearly identical among the 20 *Iris* taxa, except for minor variations in the inverted repeats. We identified 10 highly informative regions (*matK*, *ndhF*, *rpoC2*, *ycf1*, *ycf2*, *rps15-ycf*, *rpoB-trnC*, *petA-psbJ*, *ndhG-ndhI* and *psbK-trnQ*) and inferred a phylogeny from each region individually, as well as from their concatenated data. Remarkably, the phylogeny reconstructed from the concatenated data comprising three selected regions (*rpoC2*, *ycf1* and *ycf2*) exhibited the highest congruence with the phylogeny derived from the entire plastome dataset. The result suggests that this subset of data could serve as a viable alternative to the complete plastome data, especially for molecular diagnoses among closely related *Iris* taxa, and at a lower cost.

## Introduction

The genus *Iris* L. (~280 species; Iridaceae), a collection of perennial herbs, is widespread across the globe. It is the most diverse taxonomic group within the family, with a presence throughout the Northern Hemisphere [1–3]. *Iris* displays significant cytological variability, typically with x = 10 or 12 chromosomes, though some taxa exhibit varying numbers due to substantial polyploidy and aneuploidy [4, 5]. The genus also shows high morphological diversity, which can be partly explained by its diverse habitat types, from mesic to xeric [6]. Geophytic

**Funding:** This research was funded by the National Research Foundation of Korea (NRF) grant funded by the Korea government (MSIT) (No. RS-2023-00212808).

**Competing interests:** The authors have declared that no competing interests exist.

adaptations, such as rhizomes (found in subg. *Iris* and *Limniris*), bulbs (found in subg. *Xiphium*, *Scorpiris* = *Juno*, and *Hermodactylodes*), and occasionally swollen storage roots (found in subg. *Nepalensis*; [8]), are evident, particularly in xeric habitats, accompanied by slender, linear leaves and recessed stomata [1, 2, 7]. Notably, *Iris* also exhibits unique floral characteristics, such as petaloid-style branches, discrete perianth whorls, and a floral tube with basal nectary tissue [3, 7, 8]. Renowned for their exquisite beauty, Iris flowers are highly sought after as ornamental plants and have been utilized for culinary, medicinal, and horticultural purposes since ancient times, tracing back to ancient Greece [9].

*Iris* is one of the most taxonomically challenging groups. The genus comprises six subgenera and 12 sections, delineated based on characteristics such as sepal beards, crests, seed arils, and underground storage organs [7, 10–13]. However, recent molecular studies have documented discrepancies between the morphological and molecular data in terms of phylogenetic relationships [2, 6, 14, 15]. Certain taxa, grouped together due to morphological similarities, often fail to form cohesive genetic clusters in molecular phylogenies [2, 6, 14, 15]. In addition, the key morphological characteristics used for taxonomic classification lack homology or synapomorphy [2, 6, 14, 15], further complicating the taxonomy. These taxonomic difficulties are likely further complicated by frequent hybridization among congeners [16–18]. In fact, inter- and intra-specific hybridization has been strongly associated with the taxonomic issues in *Iris* [18–20]. Considering these taxonomic challenges, comprehensive and updated investigations into the phylogeny of *Iris* are required. However, genomic information that can be applied to solve these taxonomic difficulties is limited. For instance, the most recent molecular phylogenetic study on *Iris* largely depended on just a few plastid markers, including *matK*, *ndhF*, *trnL-trnF*, *trnQ-rps16*, and *trnS-trnfM*, leading to many unresolved taxonomic relationships [21].

Owing to the conserved nature of the plastid genome among angiosperms, it offers useful molecular tools for phylogenetic analyses, particularly at higher taxonomic levels [22, 23]. Starting with *rbc*L [24–26] and followed by *atpB* [27, 28], dozens of plastid regions, such as *ndh*F, *mat*K, *trn*L-*trn*F, and *trn*S-*trn*G, have been widely applied in phylogenetic studies [23, 29]. More recently, employing complete plastid genome sequences in phylogenetic studies has become feasible. In fact, plastome phylogenies have shown promising potential in resolving complex phylogenetic relationships among problematic taxa [23]. One compelling example is the phylogenetic position of *Amborella* in angiosperms, as demonstrated by Drew et al. [30]. In accordance with recent genomic advances, several studies have been conducted on the plastid genomes of *Iris*. However, the *Iris* plastomes exhibited relatively conservative sizes and structures [31, 32]. Given the significance of plastid markers with maternally inherited characteristics, identifying plastid markers with varying levels of polymorphism is of great importance to understanding the complexity of the phylogenetic relationships among *Iris* taxa.

With the advent of advanced sequencing techniques, over 90 *Iris* plastomes have recently been characterized (http://www.ncbi.nlm.nih.gov/genomes) [31, 33–36]. Genome-scale markers provide good insight into phylogenetic relationships, as observed in many plastome phylogenies [37]. In this study, we aimed to 1) determine the complete structure of the plastome of *Iris orchioides* (endemic to Central Asia [38]), 2) characterize the architecture and molecular evolution of *Iris* plastomes via comparative analysis with known *Iris* plastomes, and 3) identify appropriate plastid markers for distinguishing the phylogenetic relationships among *Iris* taxa.

## Methods

### Ethics approval and consent to participate

The plant material in this study was obtained from the wild with a legal permission for sample collection. This study protocol complies with relevant institutional, national, and international

guidelines and legislation. This study protocol also complies with the IUCN Policy Statement on Research Involving Species at Risk of Extinction and the Convention on the Trade in Endangered Species of Wild Fauna and Flora.

## Sampling, DNA isolation, and sequencing

In this study, we sequenced and assembled the complete plastid genome of *I. orchioides*. Young *I. orchioides* leaves were collected from Yangiabad, Uzbekistan (N 41°08′32.2″, E 70° 07′30.4″). The voucher specimen was prepared and deposited at the Herbarium of the Korea National Arboretum (KH) under accession number KHB1544459. We identified the species according to the key morphological characters described by Sennikov et al. [38]. The collected leaf samples were quickly dried with silica gel in a zip-lock plastic bag and stored at room temperature until further use. Genomic DNA was isolated using the DNeasy Plant Mini Kit, following the manufacturer's protocol (Qiagen, Hilden, Germany). We checked the quantity of the isolated DNA using a NanoDrop ND1000 system (Thermo Fisher Scientific, MA, USA; quality cutoff, OD 260/280 ratio between 1.7–1.9). The isolated DNA was visualized using 1% agarose gel electrophoresis. We constructed the Illumina paired-end libraries of *I. orchioides* with insert sizes of 300 bp and sent to LABGENOMICS (http://www.labgenomics.co.kr, Sungnamsi, Korea) for sequencing. The prepared libraries were sequenced on MiSeq platform (Illumina Inc., San Diego, CA, USA). We filtered poor quality reads (Phred score, Q < 20) with trim function implemented in CLC Assembly Cell package v. 4.2.1 (CLC Inc., Denmark).

## Genome assembly and annotation

The complete plastid genome of *I. orchioides* was assembled by employing the low-coverage whole-genome sequence (dnaLCW) method [39] with the CLC de novo assembler (CLC Assembly Cell package) and SOAPdenovo (SOAP package v. 1.12). We used default setting for all parameter values for the pipeline run. The Gapcloser function (SOAP package) was applied for gap filling. We also performed a reference-based genome assembly using the complete genome sequence of *I. domestica* (GenBank accession number: PP475539). For the reference-based assembly, the contigs obtained from the primary de novo assemblies were aligned to the reference plastid genome. The aligned contigs were then assembled in Geneious v. 2019.0.4 (http://www.geneious.com).

We annotated the assembled plastid genome using the Dual Organellar GenoMe Annotator (DOGMA [40]) with a few modifications for the start and stop codons. Plastid-bacterial genetic codes were used to determine the protein-coding genes. To confirm the tRNA boundaries, we scanned the tRNAs using tRNAscan-SE with the default settings [41]. The circular plastome map was visualized using OGDRAW (http://ogdraw.mpimp-golm.mpg.de/). The assembled and annotated plastid genome sequences of *I. orchioides* were deposited in GenBank (MT254070.1).

## Genome structure and comparative analysis

We compared the genome structure, size, gene content, and number of repeats across the 20 *Iris* taxa including *I. orchioides*. To simplify the comparison process among the numerous available samples (approximately 90 accessible via https://www.ncbi.nlm.nih.gov/genome/organelle/), we opted to focus on one or a few taxa at the section level. All plastid genomes, excluding that of *I. orchioides*, were downloaded from GenBank (S1 Table). The GC content was determined using Geneious software. All plastome sequences of the 20 *Iris* taxa were aligned on MAFFT (http://mafft.cbrc.jp/alignment/server/) with the default settings. The aligned sequences were then visualized using the Shuffle-LAGAN mode in mVISTA

(http://genome.lbl.gov/vista/mvista/submit.shtml). We used the complete and annotated plastome of *I. gatesii* as a reference to plot the mVISTA results. We further visualized IR boundaries using IRscope to compare among 20 *Iris* species (Amiryousefi et al. 2018). Sequence divergence (π) among the 20 *Iris* taxa was computed in DnaSP v. 6.0 [42], using a 600-bp window size and a 200-bp step size. We used CodonW (http://codonw.sourceforge.net/) to analyzed the distribution of the codon usage with RSCU ratio for all protein-coding genes.

Repeat elements were identified using the two following approaches: 1) using web-based SSR finder MISA-web (https://webblast.ipk-gatersleben.de/misa/) with varying thresholds (10 for mono-, 5 for di-, 4 for tri-, and 3 for tetra-, penta-, and hexa-nucleotide repeats); 2) REPuter was used to determine the size and type of repeats [43]. For REPuter analysis, the parameters were set as follows: minimal repeat size, 30 bp; hamming distance, 3 kb; sequence identity, ≥ 90%.

### Identifying plastid markers and testing the utility in phylogenetic inferences

We further assessed the phylogenetic informativeness (PI) of each protein coding gene and 5 Intergenic Spacer (IGS) regions using PhyDesign [44]. Plastomes typically contain over 100 Intergenic Spacer (IGS) regions, and the positions of each IGS can sometimes overlap with other genes. To simplify the analysis and avoid complexity, we opted to include only the 5 most variable Intergenic Spacer regions (IGSs) based on the π values estimated from DnaSP for the PhyDesign analysis. The HyPhy algorithm and per-site profile approach were used to calculate substitution rates per site with the default settings [45]. We inferred ML trees for 20 *Iris* taxa to identify high PI regions based on the concatenated sequence data of 79 protein coding genes and the 5 hypervariable IGSs. Subsequently, we transformed the ML tree into an ultrametric tree using the chronos function implemented in the ape R package [46]. Following the ML tree inference and ultrametric tree construction, we proceeded to estimate phylogenetic informativeness and selected the 10 most informative regions (5 genes and 5 IGSs). To assess the effectiveness of the selected regions, we conducted Maximum Likelihood (ML) tree inference for each of the 10 regions individually. Additionally, we created concatenated datasets from these 10 regions and inferred ML phylogenies to evaluate the performance of the combined dataset. In total, we generated 1,023 ML trees from the concatenated data, employing various combinations.

We reconstructed a phylogeny from complete plastome sequences of 73 *Iris* taxa with 84 accessions obtained from GenBank, along with two species of *Moraea* (Iridaceae) as outgroups (genome sizes and GenBank accession numbers are listed in S1 Table). The 86 plastome sequences were aligned using MAFFT with default settings and were then manually edited for ambiguous locations using the Geneious alignment viewer. Gaps in the sequences were treated as missing data. We inferred the phylogeny using and ML methods. To determine the best-fitting substitution model, the Akaike information criterion, implemented in jModelTest v. 2.1.10 [47], was used. The ML phylogeny was constructed using RAxML v. 8.2.4 based on the GTR GAMMA model with 1,000 rapid bootstrap replicates for the node support.

We used the plastome tree to test utility of the selected plastid regions in phylogenetic analyses. The trees inferred from the high PI regions were compared with the complete plastome phylogeny as a reference. If a tree inferred from one of the selected regions exhibited congruence with the plastome tree, we regarded the selected region as a viable alternative to the entire plastome dataset for phylogenetic reconstruction. To evaluate this congruence, we utilized TreeDist, an R script (https://github.com/ms609/TreeDist) that measures the topological differences between pairs of trees using generalized Robinson-Foulds distances, which compare bipartitions between trees.

## Results

### Plastid genome assembly and genome annotation of *Iris orchioides*

The genomic library of *I. orchioides* produced 10 million high-quality 300-bp paired-end reads. The average number of reads after initial trimming was approximately 9 million, and the average per-base coverage was 241 (Table 1). The final plastid genome of *I. orchioides* assembled in the current study showed a typical quadripartite structure divided into four regions, including a pair of inverted repeats (IRs; 25,508 bp), a large single-copy region (LSC; 82,271 bp), and a short single-copy region (SSC; 18,335 bp; Fig 1 and Table 1). The plastid genome of *I. orchioides* contained 113 genes comprising 79 protein-, 30 tRNA-, and 4 rRNA-coding genes (Table 2). Of the 113 genes, 20 genes (all four rRNA-, eight of the tRNA-, and six of the protein-coding genes, as well as two conserved ORFs) were duplicated, resulting in a total of 133 genes (Table 2). We identified 15 genes carrying a single intron and three genes (*rps12*, *ycf3*, and *clpP*) with two introns in *I. orchioides* (Table 2); one gene (*psbZ*) contained three introns (Table 2). No pseudogenes were observed in the *I. orchioides* plastid genome (Table 2).

**Table 1. Summary of the plastid genome characteristics and sample sources of the 20 *Iris* taxa used in this study.** Collection- and assembly-related information is only presented for the newly sequenced *Iris orchioides* plastome.

| Subgeneric classification | Species | NCBI accession No. | Total length (bp) | LSC length (bp) | SSC length (bp) | IRa length (bp) | IRb length (bp) | Total GC content (%) | Total number of genes |
|---|---|---|---|---|---|---|---|---|---|
| subg. *Iris* sect. *Iris* | *Iris germanica* | MZ571477 | 158,816 | 82,596 | 18,522 | 28,849 | 28,849 | 38.1% | 113 |
| subg. *Iris* sect. *Psammiris* | *Iris bloudowii* | PP069562 | 153,322 | 82,364 | 18,524 | 26,217 | 26,217 | 37.9% | 113 |
| subg. *Iris* sect. *Oncocyclus* | *Iris afghanica* | OR098702 | 153,234 | 82,517 | 18,373 | 26,172 | 26,172 | 37.9% | 113 |
| subg. *Iris* sect. *Oncocyclus* | *Iris gatesii* | KM014691 | 153,441 | 82,659 | 18,376 | 26,203 | 26,203 | 37.9% | 113 |
| subg. *Iris* sect. *Iris* (= *Pardanthopsis*) | *Iris dichotoma* | MK593157 | 153,573 | 83,071 | 18,140 | 26,181 | 26,181 | 37.9% | 113 |
| subg. *Iris* sect. *Iris* (= *Belmacanda*) | *Iris domestica* | MT001880 | 153,724 | 83,127 | 18,169 | 26,214 | 26,214 | 37.9% | 113 |
| subg. *Scorpiris* | *Iris orchioides* | PP475539 | 151,622 | 82,271 | 18,335 | 25,508 | 25,508 | 38.0% | 113 |
| subg. *Scorpiris* | *Iris hippolyti* | OK138594 | 151,171 | 81,938 | 18,307 | 25,463 | 25,463 | 38.0% | 113 |
| subg. *Scorpiris* | *Iris pseudocapnoides* | OM990831 | 151,393 | 82,228 | 18,239 | 25,463 | 25,463 | 38.0% | 113 |
| subg. *Xiphium* | *Iris rutherfordii* | OP715666 | 153,239 | 82,500 | 18,383 | 26,178 | 26,178 | 38.2% | 113 |
| subg. *Hermodactyloides* | *Iris tuberosa* | OP715674 | 153,239 | 82,500 | 18,371 | 26,184 | 26,184 | 37.9% | 113 |
| subg. *Limniris* sect. *Limniris* ser. *ensatae* | *Iris lactea* | MT740331 | 152,409 | 82,257 | 18,102 | 26,025 | 26,025 | 38.0% | 113 |
| subg. *Limniris* sect. *Limniris* ser. *Tenuifoliae* | *Iris loczyi* | MT254070 | 150,940 | 80,907 | 17,853 | 26,090 | 26,090 | 38.3% | 113 |
| subg. *Limniris* sect. *Limniris* ser. *Longipetalae* | *Iris missouriensis* | MH251636 | 153,084 | 82,405 | 18,255 | 26,212 | 26,212 | 37.9% | 113 |
| subg. *Limniris* sect. *Limniris* ser. *Laevigatae* | *Iris pseudacorus* | MK593164 | 152,562 | 82,786 | 17,880 | 25,948 | 25,948 | 37.9% | 113 |
| subg. *Limniris* sect. *Limniris* ser. *Ruthenicae* | *Iris ruthenica* | MK593167 | 152,287 | 82,311 | 18,136 | 25,920 | 25,920 | 38.2% | 113 |
| subg. *Limniris* sect. *Limniris* ser. *Sibiricae* | *Iris sanguinea* | KT626943 | 152,408 | 82,340 | 18,016 | 26,026 | 26,026 | 38.0% | 113 |
| subg. *Limniris* sect. *Limniris* ser. *Chinenses* | *Iris speculatrix* | OK274247 | 152,368 | 82,003 | 17,941 | 26,212 | 26,212 | 38.0% | 113 |
| subg. *Limniris* sect. *Lophiris* | *Iris japonica* | OK448493 | 152,443 | 83,237 | 18,490 | 25,358 | 25,358 | 37.9% | 113 |
| subg. *Limniris* sect. *Lophiris* | *Iris tectorum* | MW201731 | 153,253 | 82,833 | 18,562 | 25,929 | 25,929 | 37.9% | 113 |

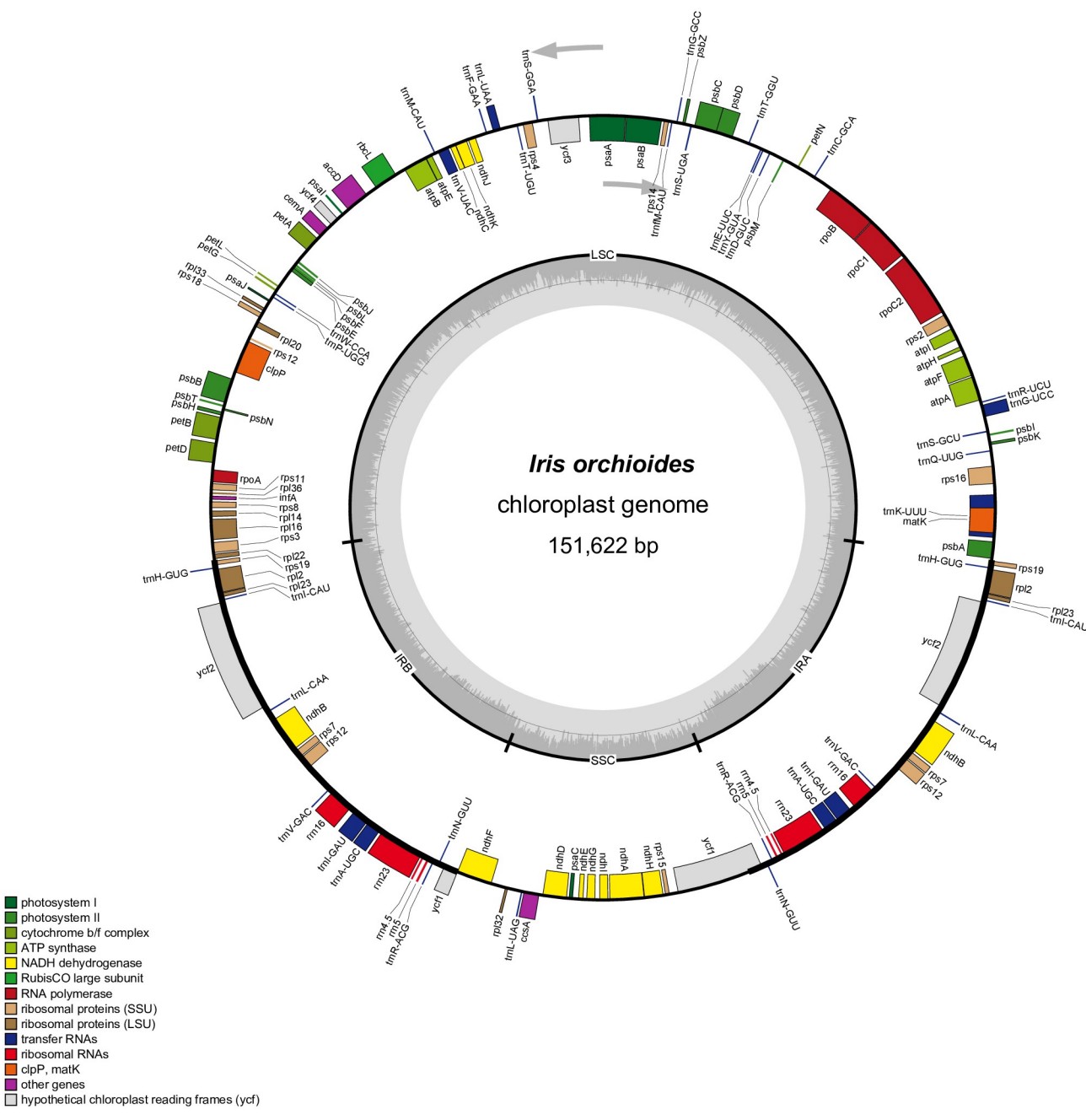

**Fig 1. Gene map of the complete plastid genome of *Iris orchioides*.** The conserved plastid genes are presented in the colored boxes. Genes located inward from the black-lined circle are transcribed clockwise, whereas the ones outside the circle are transcribed counter-clockwise. The gray bars shown in the inner circle indicate the GC content.

## Comparative analysis of plastid genome structure and polymorphism

The complete *I. orchioides* plastid genome length was 151,622 bp. The GC content was 38%, similar to that in other *Iris* species (37.9–38.3%, average = 38.0%; Table 1). The IR region was slightly shorter in three taxa of subg. *Scorpiris* [*I. orchioides* (25,508 bp), *I. hippolyti* (25,463 bp), *I. pseudocapnoides* (25,463 bp)] than in the other 17 *Iris* species (> 25,929 bp; Table 1). In

**Table 2. List of genes found in the *Iris orchioides* plastome.** ×2 indicates the genes that were duplicated in the IR regions. Asterisks refer to genes with one (*), two (**), or three introns (***).

| | Genes groups | Genes names |
|---|---|---|
| Transcription & translation | Large subunit ribosomal proteins | *rpl2*(×2)*, *rpl14*, *rpl16**, *rpl20*, *rpl22*, *rpl23*(×2), *rpl32*, *rpl33*, *rpl36* |
| | Small subunit ribosomal proteins | *rps2*, *rps3*, *rps4*, *rps7*(×2), *rps8*, *rps11*, *rps12*(×2)**, *rps14*, *rps15*, *rps16**, *rps18*, *rps19*(×2) |
| | RNA polymerase | *rpoA*, *rpoB*, *rpoC1**, *rpoC2* |
| | ribosomal RNAs | *rrn16*(×2), *rrn23*(×2), *rrn4.5*(×2), *rrn5*(×2) |
| | Transfer RNAs | *trnA-UGC*(×2)*, *trnC-GCA*, *trnD-GUC*, *trnE-UUC*, *trnF-GAA*, *trnG-GCC*, *trnG-UCC**, *trnH-GUG*(×2), *trnI-CAU*(×2), *trnI-GAU*(×2)*, *trnK-UUU**, *trnL-CAA*(×2), *trnL-UAA**, *trnL-UAG*, *trnM-CAU*, *trnN-GUU* (×2), *trnP-UGG*, *trnQ-UUG*, *trnR-ACG*(×2), *trnR-UCU*, *trnS-GCU*, *trnS-GGA*, *trnS-UGA*, *trnT-GGU*, *trnT-UGU*, *trnV-GAC*(×2), *trnV-UAC**, *trnW-CCA*, *trnY-GUA*, *trnfM-CAU* |
| Photosynthesis | Photosystem_I | *psaA*, *psaB*, *psaC*, *psaI*, *psaJ*, *ycf3***, *ycf4* |
| | Photosystem_II | *psbA*, *psbB*, *psbC*, *psbD*, *psbE*, *psbF*, *psbH*, *psbI*, *psbJ*, *psbK*, *psbL*, *psbM*, *psbN*, *psbT*, *psbZ**** |
| | NADH oxidoreductase | *ndhA**, *ndhB*(×2)*, *ndhC*, *ndhD*, *ndhE*, *ndhF*, *ndhG*, *ndhH*, *ndhI*, *ndhJ*, *ndhK* |
| | Cytochrome b6/f | *petA*, *petB**, *petD**, *petG*, *petL*, *petN* |
| | ATP synthase | *atpA*, *atpB*, *atpE*, *atpF**, *atpH*, *atpI* |
| | RUBISCO | *rbcL* |
| ATP-dependent protease subunit P | | *clpP*** |
| Other genes | Plastid envelope membrane protein | *cemA* |
| | Maturase | *matK* |
| | c-Type | *ccsA* |
| | Translation initiation factor | *infA* |
| | Subunit acetyl-CoA-carboxylate | *accD* |
| Conserved ORFs | | *ycf1*(×2), *ycf2*(×2) |

contrast, *I. germanica* had significantly large IRs (28,849 bp). The size of each quadripartite section varied among the 20 *Iris* taxa, which was mainly attributed to pronounced variation in the LSC region [80,907 (*I. loczyi*)–83,237 bp (*I. japonica*)]. Overall, we found high sequence divergence among the 20 *Iris* taxa in the non-coding regions, whereas nearly no sequence variation was observed in the untranslated regions (Fig 2). There was little variation in the sequences of the coding genes across the 20 *Iris* taxa; however, some genes, such as *ycf*1, *ycf*2, and *rpl16*, harbored relatively higher levels of sequence divergence (Fig 2). The highest sequence polymorphism was found in *I. loczyi*, particularly for *ycf*2 (Fig 2).

The sequence variability among the 20 taxa was further explored using nucleotide polymorphism ($\pi$). The mean sequence diversity was 0.018 (ranging from 0.0005–0.06; Fig 3). The SSC region showed the highest mean sequence diversity (mean $\pi = 0.031$), whereas that estimated for the two IR regions was the lowest (mean $\pi = 0.007$; Fig 3). We identified 6 hypervariable regions ($\pi \approx$ or $> 0.04$) including 5 IGS regions: *ycf1* and *rps15-ycf1* IGS ($\pi = 0.049$–0.056), *rpoB-trnC* IGS ($\pi = 0.044$), *petA-psbJ* IGS ($\pi = 0.042$), *ndhG-ndhI* IGS ($\pi = 0.04$), and *psbK-trnQ* IGS ($\pi = 0.04$; Fig 3).

The gene content organization and gene size showed a few distinctive variations, although most of the genes were structurally conserved across the 20 *Iris* taxa (Fig 4). Expansions and contractions were observed in the IR regions of the 20 *Iris* taxa (IR size = 25,463–28,849 bp; Table 1). All 20 *Iris* taxa had *trn*H-GUG in the IR region, as observed in many monocot species (Fig 4). The border between the LSC and IRb regions was between *rpl22* and *rps19*, except in *I. japonica*. The border in *I. japonica* was placed within the *rps19* gene. The distance between

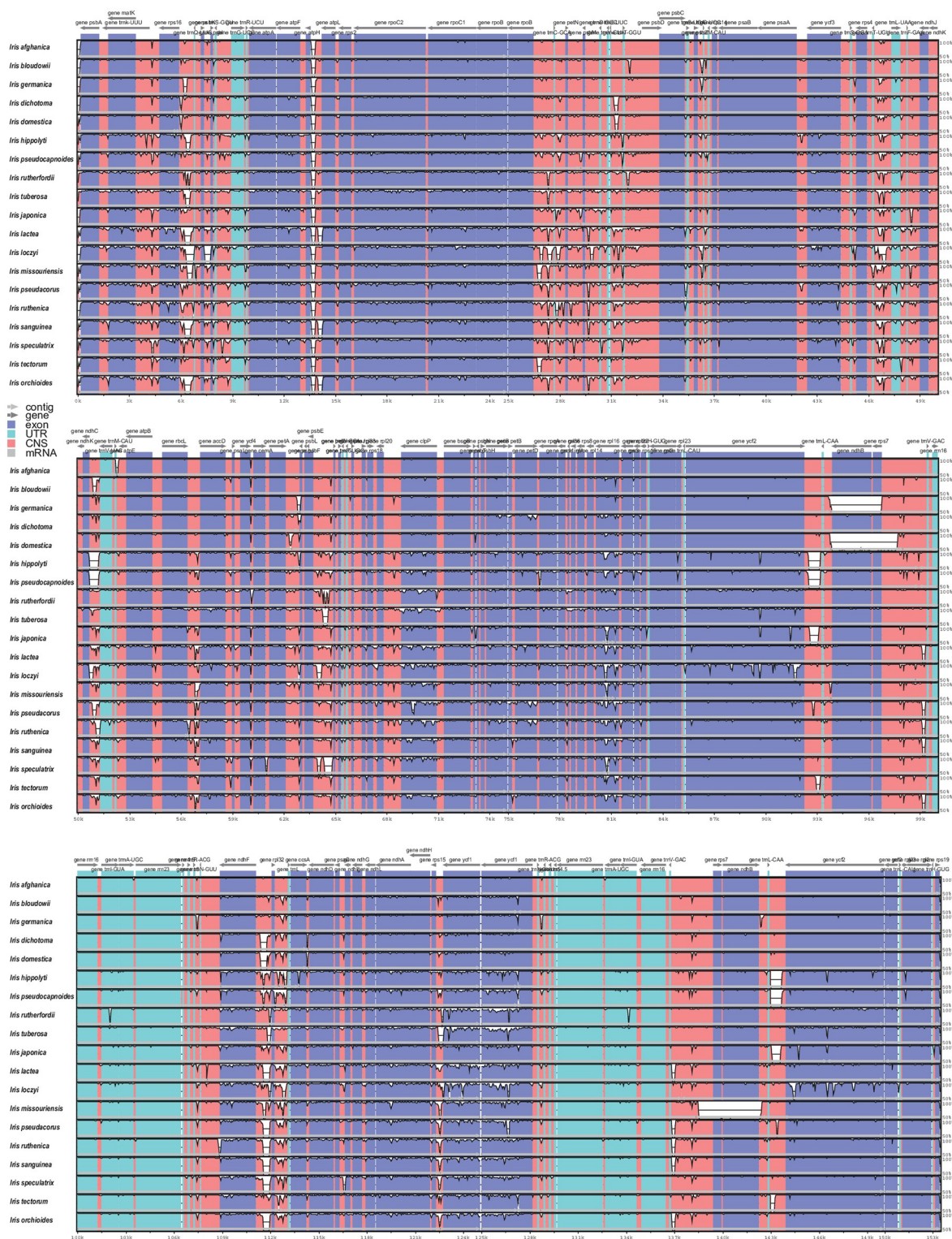

**Fig 2. mVISTA plot of the 20 *Iris* taxa.** The bar indicates the percent sequence identity of the 20 *Iris* species. *I. gatesii* was used as a reference sequence to estimate the percent sequence identities.

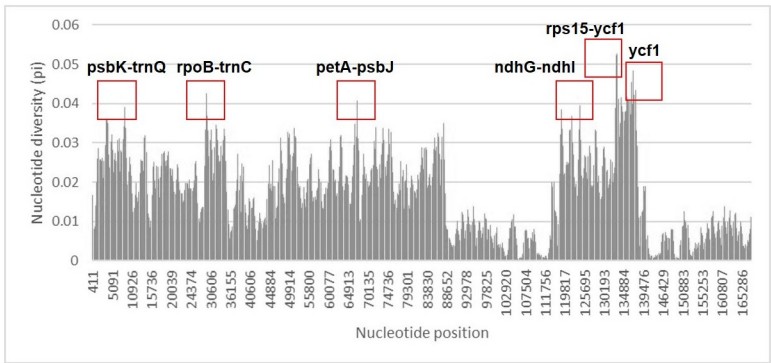

**Fig 3. Nucleotide diversity (π) plot.** The nucleotide diversity among 20 *Iris* taxa was estimated. The dashed lines are the borders of the LSC, SSC, and IR regions.

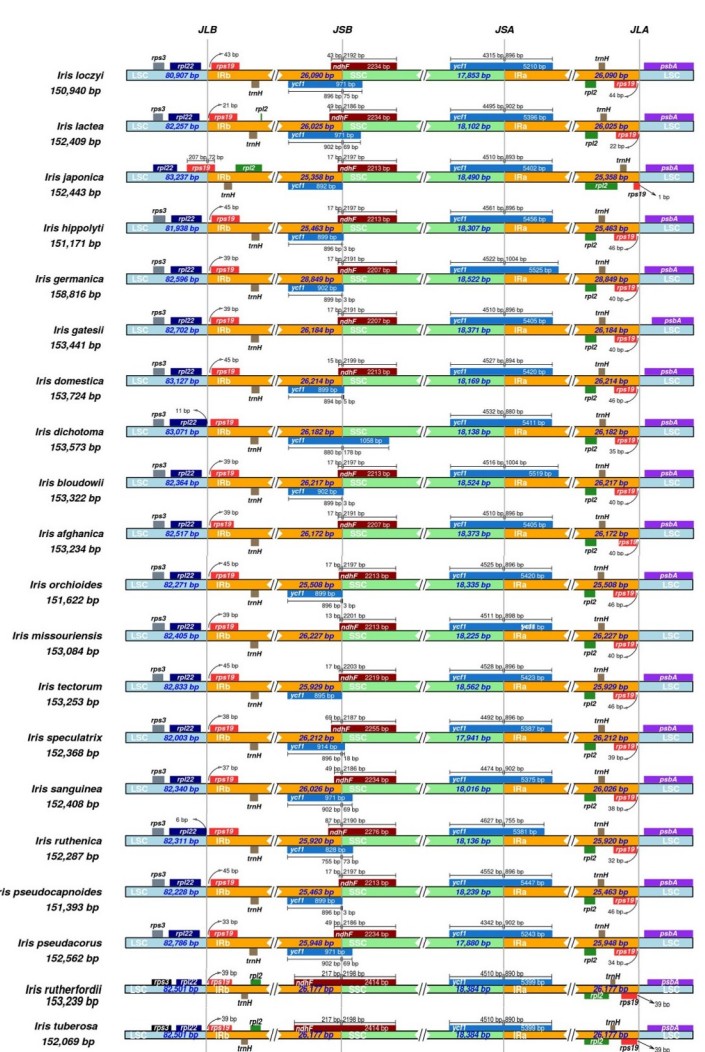

**Fig 4. Comparison of the LSC, SSC, and IR region boundaries among the plastid genomes of 20 *Iris* taxa.**

*rps19* and the border region varied from 21–45 bp (Fig 4). In *I. dicotoma*, the border between the IRb and SSC regions was located within the gene *ycf1*, showing a distinct 180-bp contraction (Fig 4). Although we observed IR region contractions and expansions, these changes did not affect the positions of the genes near the borders (Fig 4).

## Codon usage pattern

On average among the 20 *Iris* taxa, the coding sequence (CDS) length was ~79,680 bp. We found 79 genes encoded by 26,560 codons that were classified into 64 types. Of the 20 amino acids (AAs), leucine (2661–2752 encoding codons, 10.3%; S2 Table) was the most abundant, followed by isoleucine (1983–2302 encoding codons, 8.6%; S2 Table). The least abundant AA in the 20 *Iris* taxa was cysteine, encoded by 265–318 codons (1.2%; S2 Table). ATG was the initiator codon for most protein-coding genes, with three exceptions (*rps19*-GTG, *rpl2*-ACG, and *ndhD*-ACG). Relative synonymous codon usage (RSCU) did not differ significantly among the 20 *Iris* taxa (S2 Table). Two AAs, methionine (AUG) and tryptophan (UGG), were at usage equilibrium (RSCU = 1; S2 Table). The highest RSCU value was observed for codon AGA (~1.9), encoding arginine, whereas the lowest RSCU value was for codon AGC (~0.31), encoding serine. *Iris dichotoma*, *I. ruthenica*, and *I. speculatrix* had 32 codons with a RSCU > 1. In the remaining *Iris* species, 31 codons were more frequently used than expected at equilibrium (RSCU > 1). The codons mostly ended with A (~31%) or U (~37%).

## Repeats

In total, 176 simple sequence repeats (SSRs) were identified in *I. orchioides*, with a minimum repeat number of 10 (Table 3). SSR numbers varied slightly across the 20 *Iris* taxa, ranging from 167 (*I. rutherfordii*) to 203 (*I. lactea*; Table 3). We identified seven repeat motif types (mono- to hexa-nucleotide SSRs and compound SSRs; Table 3). Among all 20 *Iris* taxa, the most common repeat motif was the hexa-nucleotide SSR, whereas the least frequently observed repeat motif was the penta-nucleotide SSR (Table 3). Nearly all the mono-nucleotide repeats were composed of A or T, except for the one with C located in *ycf1* and/or in *psbK-psbI* IGS (S4 Table). Over 80% of the di-nucleotide SSRs comprised "AT," and the repeat numbers varied from 10–20 (S4 Table). The number of tri- and tetra-nucleotide SSRs showed little variation among the 20 *Iris* taxa (Table 3). The repeat numbers of the penta- and hexa-nucleotide SSRs were mostly 10 and 12, respectively; however, 15 penta- and 18 hexa-nucleotide SSRs were observed in a few taxa (S4 Table).

## Identifying plastid markers and testing the utility in phylogenetic inferences

We inferred the plastome phylogeny and compared the results with the phylogeny inferred from the marker regions selected in our study. In the plastome phylogeny, *Iris* formed a monophyletic clade distinct from the outgroup taxa (Fig 5). Overall, the phylogeny of *Iris* was consistent with that proposed by Wilson (2011), with a few exceptions. Henceforth, we interpreted the results in accordance with Wilson's taxonomic system. Taxa within *Iris* were divided into two well supported clades (Fig 5). The first clade was composed of taxa from subgenera *Limniris*, *Xiphium*, and *Hermodactyloides*, while the second clade harbored the taxa classified into the subgenera *Iris*, *Scorpiris* and *Limniris* (Fig 5; bootstrap support (BS) > 98). According to the PhyDesign results, we identified five genes with the highest phylogenetic informativeness (PI): *matK*, *ndhF*, *rpoC2*, *ycf1*, and *ycf2* (Fig 6). Additionally, in combination with these five high PI genes, the five hypervariable Intergenic Spacer (IGS) regions (*rps15-ycf*, *rpoB-trnC*,

**Table 3. Summary of simple sequence repeats (SSRs) across varying unit sizes in 20 *Iris* taxa.** c denotes compound SSRs comprising more than two adjacent SSRs.

| Species | Number of SSRs | | | | | | Total |
|---|---|---|---|---|---|---|---|
| | Unit size | | | | | | |
| | 1 | 2 | 3 | 4 | 5 | 6 | |
| *Iris germanica* | 31 | 10 | 2 | 4 | 1 | 128 | 176 |
| *Iris afghanica* | 27 | 12 | 3 | 4 | 2 | 131 | 179 |
| *Iris bloudowii* | 32 | 12 | 3 | 4 | 2 | 132 | 185 |
| *Iris gatesii* | 31 | 12 | 3 | 4 | 3 | 135 | 188 |
| *Iris dichotoma* | 32 | 9 | 2 | 4 | 1 | 129 | 177 |
| *Iris domestica* | 45 | 10 | 3 | 4 | 1 | 128 | 191 |
| *Iris orchioides* | 34 | 7 | 3 | 5 | 1 | 126 | 176 |
| *Iris hippolyti* | 22 | 10 | 4 | 4 | 1 | 144 | 185 |
| *Iris pseudocapnoides* | 25 | 14 | 3 | 5 | 4 | 131 | 182 |
| *Iris rutherfordii* | 20 | 9 | 1 | 7 | 0 | 130 | 167 |
| *Iris tuberosa* | 33 | 11 | 2 | 6 | 1 | 141 | 194 |
| *Iris japonica* | 30 | 8 | 4 | 5 | 1 | 128 | 176 |
| *Iris lactea* | 34 | 20 | 2 | 7 | 1 | 139 | 203 |
| *Iris loczyi* | 28 | 10 | 4 | 5 | 1 | 122 | 170 |
| *Iris missouriensis* | 18 | 13 | 3 | 7 | 1 | 139 | 181 |
| *Iris pseudacorus* | 33 | 9 | 3 | 5 | 2 | 126 | 178 |
| *Iris ruthenica* | 36 | 14 | 2 | 7 | 1 | 130 | 190 |
| *Iris sanguinea* | 37 | 10 | 3 | 6 | 1 | 130 | 187 |
| *Iris speculatrix* | 38 | 11 | 4 | 3 | 3 | 139 | 198 |
| *Iris tectorum* | 30 | 9 | 2 | 6 | 1 | 123 | 171 |

*petA-psbJ*, *ndhG-ndhI* and *psbK-trnQ*; Fig 3) also demonstrated high informativeness for phylogenetic analysis.

Based on the TreeDist score, identified the most effective marker for molecular diagnosis. The TreeDist result indicated that a concatenated dataset comprising a combination of three regions (*rpoC2*, *ycf1* and *ycf2*) (TreeDist = 0.88; S5 Table, S2 Fig) displayed the highest congruence score with the complete plastome phylogeny. In addition, overall tree topology of the ML phylogeny inferred from the selected data was fairly congruent with that of the plastome tree (see the dashed lines for the concordances in Fig 7). Upon closer examination, we observed minimal incongruences within major clades (refer to Fig 7). In the ML tree constructed using the selected marker, the monophyly of the genus *Iris* was strongly supported. Additionally, the two main clades identified in the complete plastome tree formed monophyletic groups with robust node supports in the ML tree of the selected data (BS > 98). Consistent with the plastome phylogeny (Fig 7), two subgenera (subg. *Iris* and *Juno*) were found to be monophyletic with strong node supports (BS = 100), while others did not exhibit nested relationships. Minor discordances were primarily observed within sect. *Oncocyclous* (indicated by the blue colored clade in Fig 7). However, notably, the bootstrap support for each node in the tree generated from the selected marker was slightly lower compared to the plastome tree (Fig 7).

## Discussion

In this study, we provided information for the whole plastid genome of *I. orchioides*, an endemic to Central Asia, and compared those of 20 *Iris* plastomes. We found that three *Iris* taxa (*I. orchioides*, *I. hippolyti*, and *I. pseudocapnoides*) showed larger structural and size

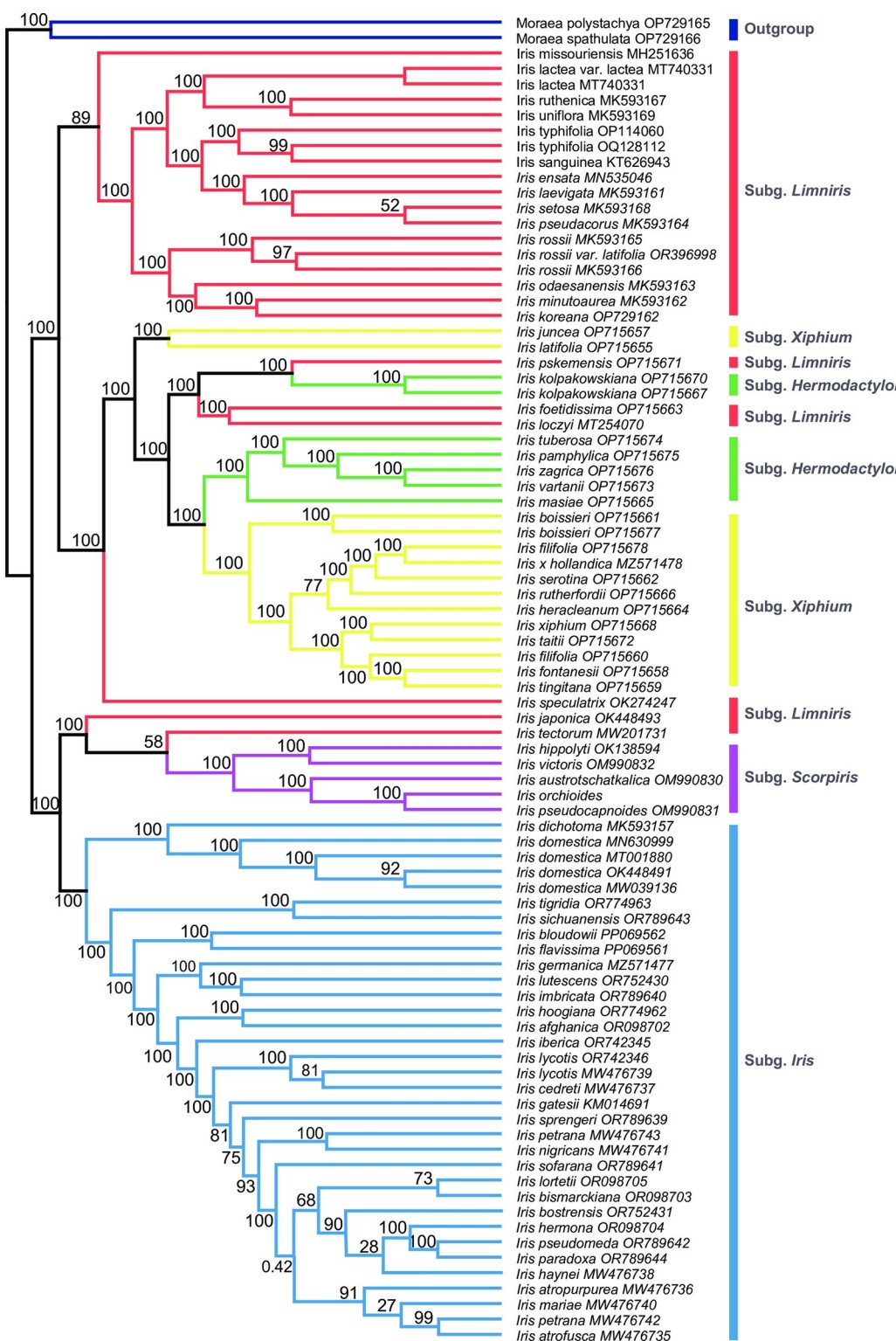

**Fig 5. Phylogenetic relationships among 84 accessions of *Iris* inferred using maximum likelihood (ML) methods based on whole-plastid genomes.** The values presented on each node indicate the bootstrap support.

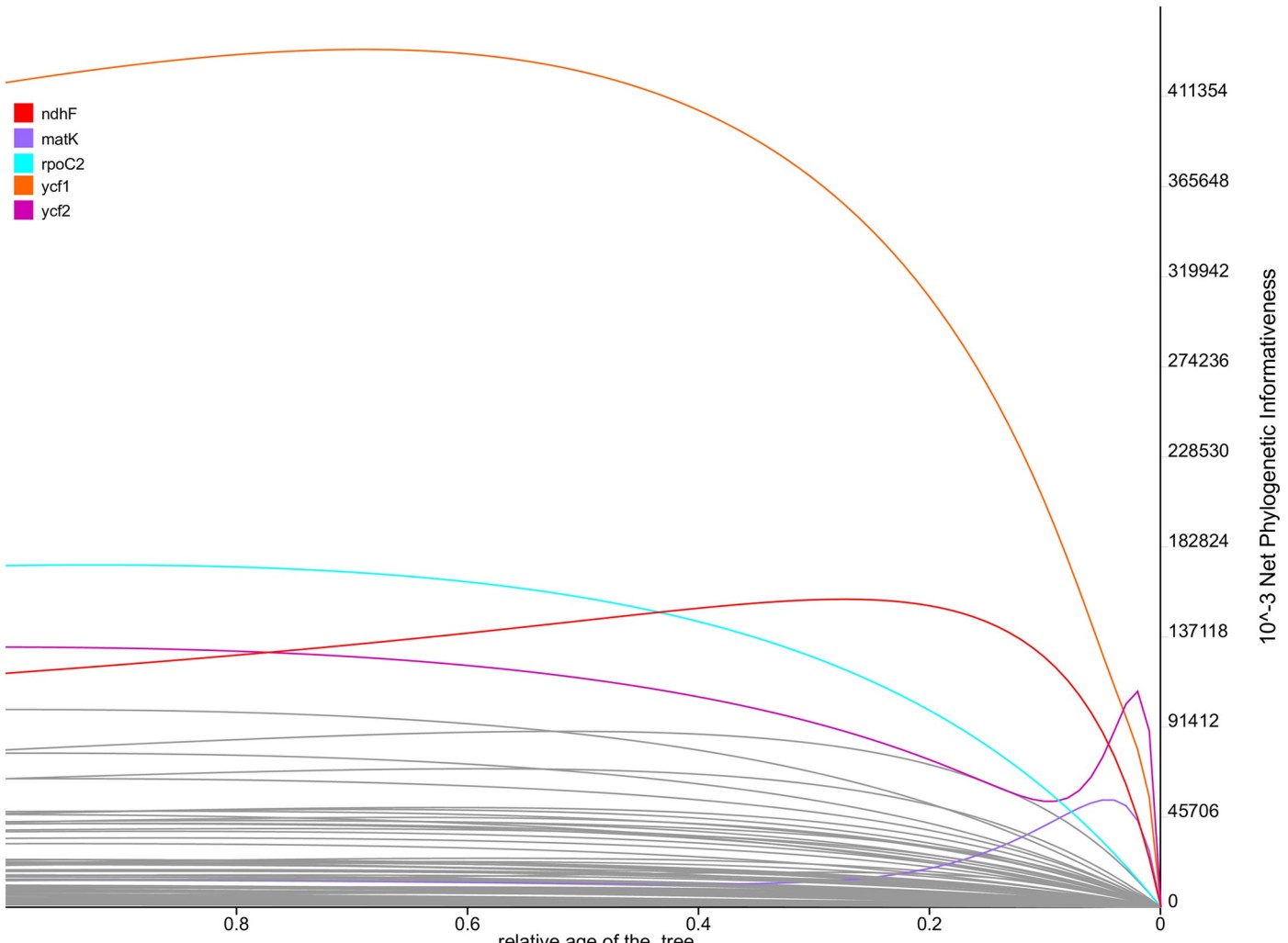

**Fig 6. Phylogenetic informativeness profiles of 79 coding sequences of *Iris* estimated in PhyDesign.** The five most informative regions are marked in color.

variations than the other *Iris* taxa. We isolated molecular markers, such as SSRs and regions with high polymorphism, that could potentially be used to study *Iris* population genetics (Fig 3). More importantly, we pinpointed 5 genes (*matK*, *ndhF*, *rpoC2*, *ycf1*, and *ycf2*) with high PI values and 5 hypervariable IGS regions (*rps15-ycf*, *rpoB-trnC*, *petA-psbJ*, *ndhG-ndhI IGS*, and *psbK-trnQ*) in *Iris*, offering valuable tools for phylogenetic analysis (see Fig 6).

Like most angiosperm plastomes, the 20 *Iris* plastomes showed relatively conserved genomic structures, characterized by typical quadripartite structures [48]. The plastome size ranged from 120–170 kbp, falling within the expected size range for angiosperms [48]. Plastid genome structure primarily depends on the organization of IRs, as their size and arrangement often vary [49, 50]. Our comparative analysis revealed size and structural variations in the *Iris* plastomes, particularly in the IR size (Figs 2 and 4). According to the mVISTA result, five species (*I. orchioides*, *I. japonica*, *I. hippolyti*, *I. tectorum*, and *I. pseudocapnoides*) show a 500-bp deletion in the intergenic region around the *ycf2* gene (Fig 4). Notably, these five species formed a monophyletic clade in the inferred plastome phylogeny (Fig 5). The *ycf2* gene also showed a high PI values likely stemming from the notable size variation observed in this gene (Fig 6).

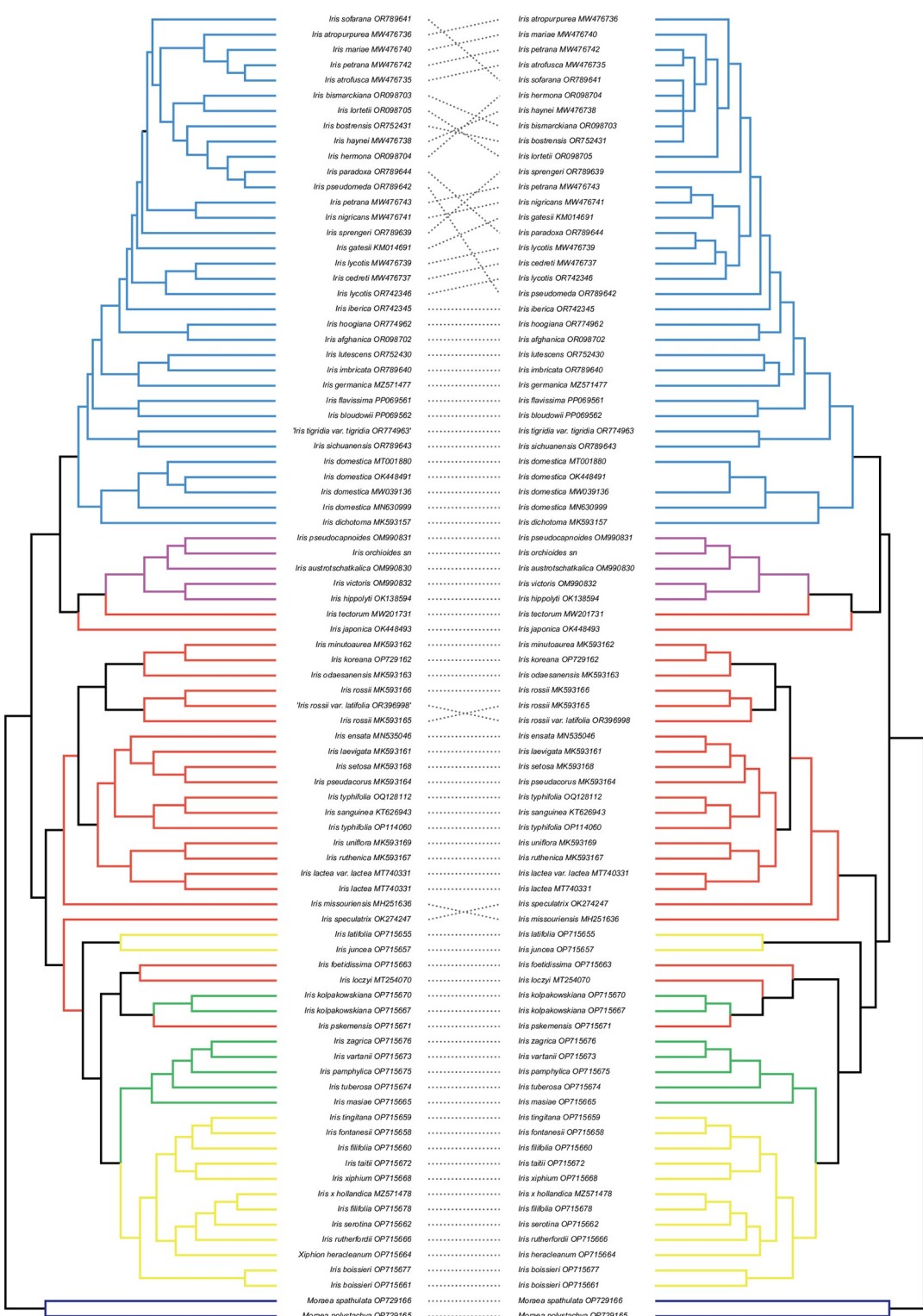

**Fig 7. Phylogenetic relationships among 84 *Iris* accessions, inferred using maximum likelihood (ML) methods based on the whole plastome (left) and the best diagnostic marker, the concatenated data with the three high PI genes (*rpoC2*, *ycf1*, *ycf2*) (right).** Dashed lines connect the same samples in both data set. Colors in the trees indicate a subgenus to which the clade belong. Blue- subg. *Iris*, Red-subg. *Limniris*, Purple- subg. *Scorpiris*, Green- subg. *Hermodactyloides*, Yellow- subg. *Xiphium*.

Accordingly, this sequence deletion might contribute strongly to the close phylogenetic relationships among the five species, highlighting the efficacy of the gene as a phylogenetic marker in *Iris*. With a few exceptions, such as tobacco (171 kbp) and geraniums (217 kbp), length variations in complete plastid genomes are not common in angiosperms [23, 51]. However, similar to our observations, previous studies have reported large sequence variations in *ycf*2 genes and their neighboring regions [52, 53]. Unfortunately, the mechanism driving this prominent structural variation in *ycf*2 is still under investigation. Regardless of the causal mechanism, the variation is worth investigating in all *Iris* taxa, as such large deletions could help to avoid the homoplasy-related challenges in *Iris* phylogeny [54].

Although the divergence of plastome sequences varies among taxa, coding regions are generally more conserved than non-coding regions, e.g., introns and intergenic spacers [55]. Likewise, in the study, the sequence divergence was lower in the coding regions (mean π of coding sequences = 0.019) than in the non-coding regions (0.021). Overall, the sequence divergences were larger in the 20 *Iris* plastomes than those observed in other taxa (average π = 0.009 in three *Papaver* species, average π = 0.003 in three *Cardiocrinum* species, and average π = 0.0007 in six *Hosta* species) [56–58]. High sequence divergence was expected, considering the taxonomic characteristics of *Iris*. The genus *Iris* is one of the most heterogeneous genera and is endowed with several infrageneric groups showing high morphological and molecular variations, e.g., subg. *Scorpiris* or *Belmacanda*. A recent study [15] divided the genus into 23 different genera based on the high molecular variation found in 10 plastid genome regions. Similaly, we isolated six hypervariable sites located in the LSC and SSC regions (*rpo*B-*trn*C, *pet*A-*psb*J, and *ycf*1; Fig 3) that can be employed to elucidate shallow-level phylogenies among closely related taxa or in population genetic studies. Sequence alignment (S1 Fig) revealed that *I. loczyi* had the largest number of polymorphic sites in *ycf*1, the gene with the highest sequence divergence (π = 0.05; S3 Table). The *ycf*1 gene encodes a protein crucial for plant viability. Ironically, despite its essential function, the gene exhibits high polymorphism and proves to be useful in both shallow and deep phylogenies [59, 60]. Our data analysis found that *ycf*1 in *I. loczyi* exhibited two large deletions (~100 bp; S1 Fig). Collectively, this gene could be an excellent marker for identification and phylogenetic inferences in closely related *Iris* taxa. Variations such as the *ycf*1 indel should be further investigated using more *Iris* taxa to identify variations that could shed light on species and population diversification in the genus.

Codon assignment bias over individual codons is a predominant phenomenon in all living taxa [61]. The two primary factors characterizing codon usage patterns are genome composition and selection toward increased translation efficiency [62, 63]. We observed significant bias toward AU (69% of all codons) in the 20 *Iris* plastid genomes, which is consistent with previous reports on plastid genomes (Campbell and Gori, 1990). In general, the plastid genomes were extremely AT-rich and relatively GC-limited (30–40% of total sequences). It is likely that the primary cause of codon usage bias observed in the 20 *Iris* plastid genomes was the AU-rich compositional bias.

We isolated 167–203 SSRs from the 20 *Iris* taxa, which is rather higher than the numbers previously reported in angiosperm species (105 SSRs in *Betula*, [64] 130 in *Paris*, [65] 50 in *Chenopodium*, [66] 250 in *Aconitum*, [67] and 48 in *Fagopyrum* [68]). Given the plethora, neutrality, and high variability, SSRs are the most frequently employed molecular markers in population genetics [69]. Regardless of utility, identifying applicable SSRs is a costly and time-consuming process [70, 71]. Over 1,000 nuclear SSR markers in *Iris* have been developed [72], although the number of SSRs that can be applied may differ across the targeted taxa, owing to varying polymorphism and amplification levels. However, maternally inherited plastid markers applicable for population-level genetic studies are scarce. The plastid SSR markers we

identified in this study offer a useful molecular tool to investigate genetic patterns of maternal inheritance at the population level. In future studies, the SSR markers newly documented in our study should further be tested for rate of polymorphism at population level with many genotype samples to determine the applicability.

*Iris* is one of the most notorious groups for taxonomic complications owing to the large morphological variation and frequent gene flow among congeners [7, 12, 13, 16, 18, 19]. Numerous classification schemes have been proposed for approximately 280 *Iris* taxa [7, 11– 13, 16, 18, 19, 21], with many relying on floristic characteristics such as sepal beards, crests, seed arils, and subterranean organs, alongside a limited number of plastid markers [7, 21]. However, despite these efforts, the taxonomy of *Iris* remains elusive, particularly when considering commercial taxa [38]. To address these taxonomic challenges, comprehensive sampling of diverse molecular markers is essential.

When inferring a phylogeny of targeted taxa, selecting appropriate molecular markers is of great concern, as the selected markers can strongly affect overall topology and divergence time estimates [73]. Recently, phylogenetic informativeness (PI), which quantitatively predicts phylogenetic signal based on estimated mutation rates, has gained prominence and is commonly employed in phylogenetic studies across various taxonomic groups [44, 74–76]. We estimated PI and identified the 10 most highly informative regions. Among them, *rpoC2* and *ycf1* stood out, offering improved overall tree topology with higher node support compared to other high PI regions (Fig 7). In fact, *rpoC2* and *ycf1* have recently been recognized for their phylogenetic utility among angiosperms [77]. Accordingly, the *rpoC2* and *ycf1* genes hold promise as useful tools for inferring *Iris* phylogeny.

In our analysis, we systematically tested various combinations involving 5 high PI genes and 5 hypervariable IGSs to identify the optimal combination for the phylogeny of the 84 *Iris* taxa. Remarkably, our selection process resulted in selecting the concatenated dataset of three regions (*rpoC2, ycf1* and *ycf2*; Fig 7, S5 Table). The phylogeny inferred using the selected dataset remained largely consistent with the whole plastome phylogeny, and the node support for most clades was robust. Section *Oncocyclus* exhibited notable incongruence between the phylogeny of the selected data and that of the complete plastome. Taxa within this section are known for their high morphological variability, and certain species have been found to be non-monophyletic in previous studies [2]. Additionally, subgenus *Xiphium* was found to be non-monophyletic in both the plastome tree and the chosen combination tree (Fig 7). This finding is consistent with recent studies indicating that subgenus *Xiphium* remained unresolved in plastome data [78]. The results suggest that concatenating the three high PI regions as phylogenetic markers can serve as a cost-effective alternative to sequencing the entire plastome. Applying whole-plastome data for phylogenetic analysis may not always be the most efficient approach, as it can be costly. Moreover, relying solely on whole-plastome phylogeny may not consistently provide the best resolution due to certain limitations inherent to this approach. The over-representation of regions with high variation in whole-plastome data can potentially introduce errors when inferring phylogeny [79, 80]. By employing the high PI regions as markers without the need to sequence the entire plastid genome, we can potentially address phylogenetic complexities and enhance molecular diagnoses between closely related Iris taxa at a reduced cost.

## Supporting information

**S1 Fig. Sequence alignment of the ycf1 region across 20 Iriss species.**
(JPG)

**S2 Fig. Phylogenetic relationships among 84 accessions of *Iris* inferred using maximum likelihood (ML) methods based on each 10 high PI regions.**
(PDF)

**S1 Table. The summary of the plastid genome sequences downloaded from GenBank for phylogenetic analysis.**
(XLSX)

**S2 Table. Codon usage pattern of 20 Iris species.**
(XLSX)

**S3 Table. Sequence divergence ($\pi$) among the 20 Iris species.**
(XLSX)

**S4 Table. Simple sequence repeats (SSRs) across varying unit sizes in 20 *Iris* species.**
(XLSX)

**S5 Table. TreeDist distances among concatenated 10 high PI regions.**
(XLSX)

## Acknowledgments

We thank Dr. K. Sh. Tojibaev for their help in collecting samples and in formal identification.

## Author Contributions

**Conceptualization:** Tae-Young Choi, Soo-Rang Lee.

**Data curation:** Tae-Young Choi.

**Formal analysis:** Tae-Young Choi.

**Funding acquisition:** Soo-Rang Lee.

**Investigation:** Tae-Young Choi, Soo-Rang Lee.

**Methodology:** Soo-Rang Lee.

**Project administration:** Soo-Rang Lee.

**Supervision:** Soo-Rang Lee.

**Validation:** Tae-Young Choi, Soo-Rang Lee.

**Visualization:** Tae-Young Choi.

**Writing – original draft:** Tae-Young Choi, Soo-Rang Lee.

**Writing – review & editing:** Soo-Rang Lee.

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
