## [Decision Letter · Decision Letter 0]

23 Jan 2024

PONE-D-23-41497Complete chloroplast genome of Irisorchioides and comparative analysis with 13 Iris plastomesPLOS ONE

Dear Dr. Lee,

Thank you for submitting your manuscript to PLOS ONE. After careful consideration, we feel that it has merit but does not fully meet PLOS ONE’s publication criteria as it currently stands. Therefore, we invite you to submit a revised version of the manuscript that addresses the points raised during the review process.

Both reviewers questioned the reason for selecting 13 additional plastid genomes among the 40 reported within the genus* Iris*. Please provide theoretical background for this decision. Additionally, parameters for the selection of 5 genes for further use in phylogenetic analyses in the genus *Iris* are not clearly presented. Some disagreements have also been noted by both reviewers between the M&M and the Results sections. The authors should put additional effort to clarify these inconsistences and to properly discuss them in the Discussion section.

Please use the term "plastid" instead of "chloroplast" and "CP" throughout the text (including the main title), because various plastid forms (chloroplasts, chromoplasts, leucoplasts, amyloplasts, and elaioplasts) contain the same genome as chloroplasts and from the section "Sampling, DNA isolation, and sequencing", I see the authors did not isolated chloroplasts from the other plastids prior to DNA extraction.

Abstract: I could not understand the statement "genomic information for the *Iris* chloroplast is limited". Can you please rephrase it to clarify its meaning. In the next sentence, the phrase "of the* Iris* chloroplast" implies that all *Iris* species contain the same chloroplast. Would "of plastids within the genus* Iris*" be better? In the same sentence, please shorten the genus name in "*Iris orchioides*" as "*I. orchioides*".

Further in Abstract: what "five selected sites" is related to? Where are these sites? Can you be more precise? In the next sentence, "five identified genes" does not tell much. First, how were they identified? Maybe selected? And, of course, please list the selected genes and criteriums for their selection here.

Introduction: Please italicize "Juno", as it presents a synonym for the genus.

We look forward to receiving your revised manuscript.

Kind regards,

Branislav T. Šiler, Ph.D.

Academic Editor

PLOS ONE

 [This research was funded by the National Research Foundation of Korea (NRF) grant funded by the Korea government (MSIT) (No. RS-2023-00212808). ].  

5. Please include the reference section of your manuscript

Reviewers' comments:

Reviewer's Responses to Questions

**Comments to the Author**

1. Is the manuscript technically sound, and do the data support the conclusions?

Reviewer #1: Partly

Reviewer #2: Yes

2. Has the statistical analysis been performed appropriately and rigorously? 

Reviewer #1: Yes

Reviewer #2: N/A

3. Have the authors made all data underlying the findings in their manuscript fully available?

Reviewer #1: Yes

Reviewer #2: Yes

4. Is the manuscript presented in an intelligible fashion and written in standard English?

Reviewer #1: No

Reviewer #2: Yes

5. Review Comments to the Author

Reviewer #1: The researchers successfully assembled the chloroplast genome of Iris orchioides and conducted comparative genomic analysis with the chloroplast genomes of 13 previously published Iris species. They identified highly polymorphic gene regions within the Iris chloroplast genome, utilizing these as markers to identify the phylogeny of Iris. The data analysis appears to be robust, lending substantial support to the majority of the conclusions drawn in this study. However, the study still presents several issues.

The key issues identified in the study are:

1. The paper states that 40 chloroplast genome sequences of Iris species are available. However, the reason for selecting 13 of these for comparative analysis with the I. orchioides chloroplast genome sequence is not clear. The text should explicitly justify this selection.

2. The interpretation of the constructed phylogenetic tree is missing, as is a discussion on how these results contribute to differentiating Iris species. Furthermore, there is no comparison of these findings with prior phylogenetic analyses based on morphology and nuclear genes.

3. The absence of population-level chloroplast genome data precludes inferring the suitability of the identified markers for population data.

4. The methodology for narrowing down the ten regions with high Pi values to five candidate sites remains unclear. Detailed explanation in the text is required.

5. The conclusion drawn from Figure 7, suggesting that the ten genes approximated the entire chloroplast genome in inferring the phylogeny of Iris, requires further clarification and evidence.

The minor issues identified in the study are:

1. Numerous textual errors are present notably in the method section (specific line numbers not provided) and line 182.

2. Figure 1 lacks an indication of the transcription direction.

3. The font of Table 1 should be modified.

4. The identification of outgroups in Figures 5 and 6 is unclear, and the significance of the various colors used is not delineated.

5. In Figure 2, areas of high polymorphism should be emphasized to enhance visibility for readers.

6. The manuscript’s English language quality requires substantial improvement.

Reviewer #2: In this study, the authors newly sequenced the cp genome of Iris orchioides and comparatively analyzed the cp genomes of other Iris species. This study had a scientific significance. However, some questions need to be revised as follows:

1. In the introduction section, S, and some groups fail to form monophyletic groups in most molecular phylogenies?? The phylogenetic relationships were not solved using some molecular markers? Please clarify it in detail. The classification of genus Iris should be detailed, and the information of subgenus was mentioned in the main text.

2. The cp genomes of 40 Iris species were released. Why only 13 species were used to compare in this study?

3. In the method section, the results of phylogenetic analysis were not consistent with the description of the method.

4. In the table 1, the classification of each species had better to be added.

5. In the results section, the description of the figure 6 and 7 was not consistent with the results. In addition, figure 5 showed the phylogenetic tree based on the cp genome data. However, another tree had also included this tree. Figure 5 should be deleted.

6. The author selected ten regions with high-pi values, their phylogenetic tree should be also shown in the text.

7. What method was used to select the five regions? Whether was the five markers suit for these analyzed species? How about other species?

8. Where was the Table S5?

9. The title of axis should be added in figure 7.

6. PLOS authors have the option to publish the peer review history of their article (what does this mean?). If published, this will include your full peer review and any attached files.

Reviewer #1: No

Reviewer #2: No

---

## [Author Response · Author response to Decision Letter 0]

13 Mar 2024

We appreciate the insightful comments from anonymous reviewers and the handling editor. The manuscript has been improved based on the comments. To navigate the updates in the revised manuscript, please use the review function in MS word. We made the changes available using the track and change function in MS word.

---

## [Editor Report · Decision Letter 1]

15 Mar 2024

Complete plastid genome of Iris orchioides and comparative analysis with 19 Iris plastomes

PONE-D-23-41497R1

Dear Dr. Lee,

We’re pleased to inform you that your manuscript has been judged scientifically suitable for publication and will be formally accepted for publication once it meets all outstanding technical requirements.

Kind regards,

Branislav T. Šiler, Ph.D.

Academic Editor

PLOS ONE
---

## [Editor Report · Acceptance letter]

24 Mar 2024

PONE-D-23-41497R1 

PLOS ONE

Dear Dr. Lee, 

I'm pleased to inform you that your manuscript has been deemed suitable for publication in PLOS ONE. Congratulations! Your manuscript is now being handed over to our production team.

Kind regards, 

on behalf of

Dr. Branislav T. Šiler 

Academic Editor

PLOS ONE